# Improving the Anchorage in Textile Reinforced Cement Composites by 3D Spacer Connections: Experimental Study of Flexural and Cracking Behaviors

**Michael El Kadi \*** , **Danny Van Hemelrijck and Tine Tysmans**

Department Mechanics of Materials and Constructions (MeMC), Vrije Universiteit Brussel (VUB), Pleinlaan 2, 1050 Brussels, Belgium
* Correspondence: michael.el.kadi@vub.be

**Abstract:** Textile-reinforced cement (TRC) composites can lead to significant material (and dimensional) savings compared to steel-reinforced concrete, particularly when applied in thin-walled structures such as façade panels, shells, etc. In conditions where the geometrical restrictions do not allow for sufficient anchorage, however, the exploitation of this reinforcement may be suboptimal and the TRC's mechanical properties may decrease. As shown in the literature, the use of 3D textile reinforcement can lead to an improved anchorage in the reinforcement points and superior post-cracking behavior in terms of bending. The question remains as to whether similar improvements can be achieved using 3D spacer connections, inserted post-manufacturing of the textiles. Therefore, this research experimentally investigated the effect of discretely inserted spacer connections on the flexural properties and cracking behavior of TRCs. Six different TRC beam configurations—varying in the placement of the spacer connections over the span—were investigated. Moreover, a comparison was made with two additional configurations: one equivalent 2D TRC system (using the same in-plane textiles but without through-thickness connections) and one 3D TRC system using knitted 3D textiles (with spacer yarns uniformly distributed). The four-point bending tests were monitored via digital image correlation (DIC) to visualize the full-field cracking pattern. The experimental results showed that the spacer connections could strongly improve the post-cracking bending stiffness and the modulus of rupture (MOR) when placed close to the free end of the sample and could also lead to reduced crack widths when placed around the midspan.

**Keywords:** 3D textile; digital image correlation; spacer; textile-reinforced concrete (TRC); transversal connection

## 1. Introduction

Today's innovations in construction are driven by the need for a sustainable built environment and for developments that reduce the use of primary resources. Textile-reinforced cement (TRC) composites combine a cementitious mortar matrix with textile reinforcement. The matrix serves as the backbone of the composite, while the textiles exhibit a ductile and controlled post-cracking tensile behavior [1–3]. The use of TRCs can reduce the use of concrete by up to 85%, compared to steel-reinforced concrete [4]. Several applications using TRCs are reported in the scientific literature—from ventilated façade panels [5] and repair and strengthening realizations [6–11] to load-bearing applications, such as shell structures [12–15], sandwich panels [16,17], and pedestrian bridges [18]—and demonstrate the material's potential for the realization of lightweight, load-bearing structures.

The use of three-dimensional textiles as reinforcement for TRCs offers significant advantages compared to traditional, planar textile reinforcement, such as superior toughness [19] and optimal fiber exploitation in flexural applications [20]. From a manufacturing perspective, the use of a singular, 3D textile reinforcement entity that can be inserted in a mold, with the concrete poured over, strongly aligns with current production practices

for concrete. From a mechanical point of view, different studies [20–23] have shown that the presence of transversal connections in 3D TRCs can lead to superior flexural post-cracking behavior of the material. Amzaleg et al. [20] have studied TRCs containing aramid and polyester transversal connections in different contents and reported the improved mechanical performance of 3D textiles, mainly when impregnated in epoxy. El Kadi et al. [22,23] have reported a post-cracking bending stiffness increase of up to three times in configurations containing 3D textiles compared to equivalent 2D configurations (without transversal connections).

Three-dimensional textiles are usually manufactured by means of a combined weaving–knitting process, during which both the in-plane textiles and the transversal connections are inserted gradually [24,25]. Alternatively, individual spacer connections can be inserted in the transverse direction between pre-manufactured, planar textiles. These spacer connections are usually larger than integrated, knitted connections, and are positioned at discrete locations rather than uniformly distributed over the textile surface. Hence, these spacers induce significant discontinuities in the matrix material. It is currently unknown whether these individual spacer connections can achieve any level of anchorage improvement.

This research, therefore, investigated the effect of transversal spacer connections on the flexural behavior of TRCs, more specifically at the level of the post-cracking bending stiffness, the modulus of rupture (MOR), and the crack formation. For this purpose, four-point bending tests were performed on eight different TRC configurations: (i) C0, an equivalent 2D configuration (containing in-plane textiles without through-thickness connections) that served as a reference; (ii) KNIT, a knitted 3D configuration (with homogeneously distributed transversal connections) that served as a benchmark for integrated through-thickness connections; (iii) C4MOMENT, a configuration with four spacer connections, placed in the moment zone of the sample (between the loading pins); (iv) C4SHEAR, a configuration with a total of four spacer connections, distributed over the shear region of the sample; (v) C6SHEAR, a configuration with a total of six spacer connections, distributed over the shear region of the sample; (vi) C4FREE, a configuration with a total of four spacer connections, distributed over the free region of the sample (outside the supported span); (vii) C16FREE, a configuration with a total of sixteen spacer connections, distributed over the free region of the sample; (viii) C28, a configuration with 28 spacer connections, distributed over the entire length of the sample. This comparison gave the first insights into what was, at that point, the unknown influence of discretely inserted spacers on the flexural behavior of TRC composites. Six samples were manufactured and tested for each configuration, resulting in a total of 48 tested samples. The macro-mechanical response, as well as the crack formation, were selected as representative of the quality of the anchorage and were both investigated in this study. The crack formation and propagation were assessed using the optical digital image correlation (DIC) technique [26].

## 2. Materials and Methods

### 2.1. Materials Selection and Reinforcement Configurations

#### 2.1.1. Textiles

The textile reinforcement was an SBR-coated, 3D glass textile with knitted transversal connections (height 8.5 mm) made of polyester (Figure 1a). However, for all configurations except the KNIT, these knitted connections were removed in order to create the 2D textiles (Figure 1b). As such, the in-plane textile reinforcement layers are equal for all configurations and have a mesh size of 22.5 mm × 22.5 mm (see Table 1 for all properties).

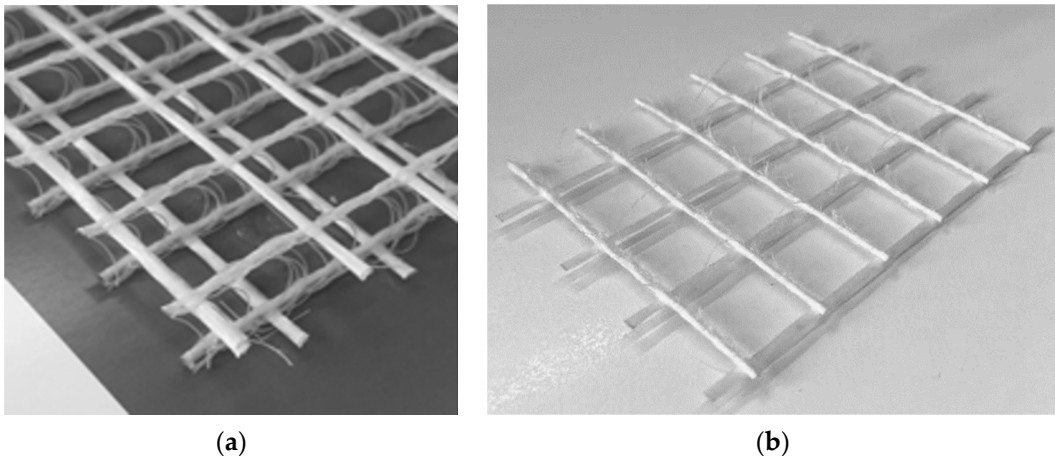

|     |     |
| :-: | :-: |
| (**a**) | (**b**) |

**Figure 1.** Images of: (**a**) the 3D glass textile; (**b**) the 2D equivalent textile.

**Table 1.** Properties of the textile reinforcement, obtained from [27].

| Material | Grid Size (mm × mm) | Textile Density Coated (Both Layers) (g/m$^2$) | Textile Density Uncoated (Both Layers) (g/m$^2$) | Coating | Transversal Connection |
| --- | --- | --- | --- | --- | --- |
| AR Glass 2400tex | 22.5 × 22.5 | 614 | 536 | SBR | Polyester PET |

### 2.1.2. Connections

Apart from the knitted transversal connections (Figure 1) used for the KNIT configuration, spacer connections were used to achieve the other six spacer configurations (Figures 2 and 3) [28]. These HF840MO polypropylene connections were inserted manually at different positions to achieve configurations with the lever arm (the same as in the KNIT configuration) between the individual textile layers.

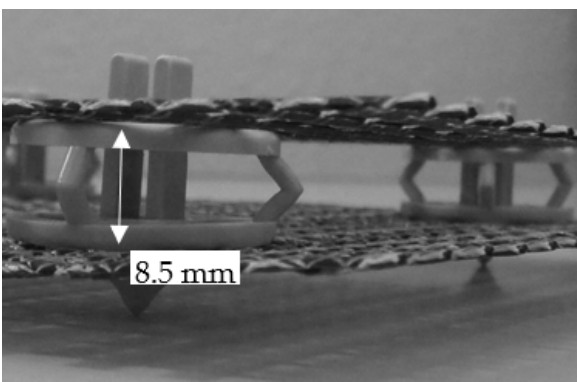

**Figure 2.** The spacer connections, inserted between two textile layers.

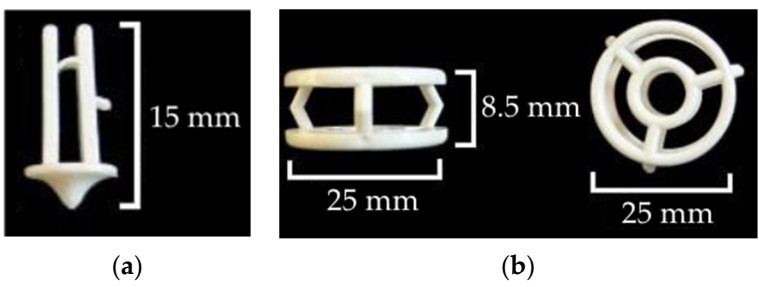

|     |     |
| :-: | :-: |
| (**a**) | (**b**) |

**Figure 3.** Dimensions of the spacer connections: (**a**) the central pin; (**b**) the circular piece.

### 2.1.3. Geometry and Spacer Configurations

An overview of all configurations, the location of the spacer connections, and the specimen dimensions are shown in Figure 4. The displayed grid corresponds to the actual mesh of the planar textiles. The location of the two loading pins and of the supports are indicated with a full black line.

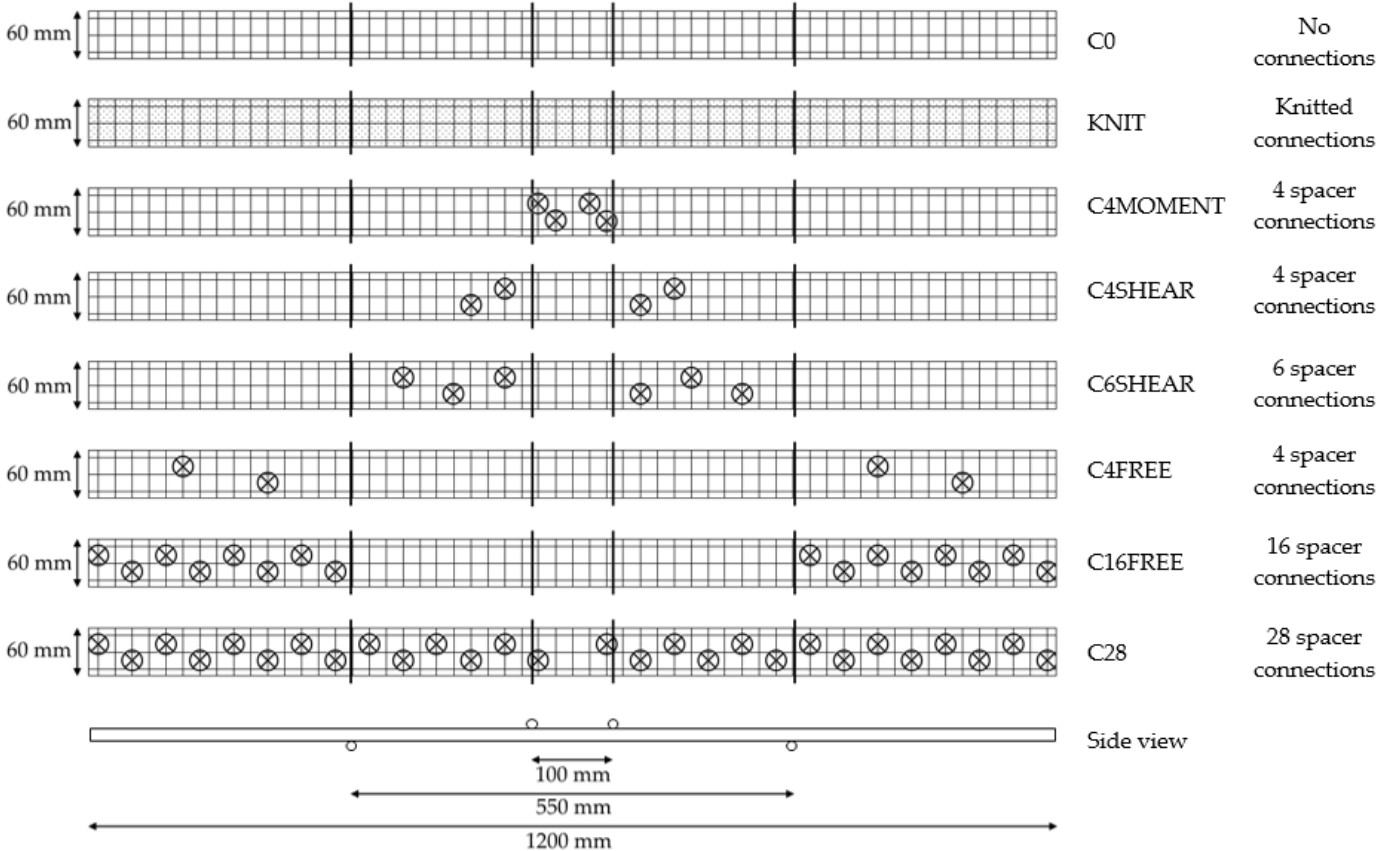

**Figure 4.** Configurations of the considered TRCs, indicating the location of the spacers and the loading setup.

All specimens had a concrete cover thickness of 2 mm at the top and bottom, two textile layers of 1.25 mm thickness, and a lever arm of 8.5 mm between the layers, resulting in a total thickness of 15 mm. The specimen length and width were 1200 mm and 60 mm, respectively. The fiber volume fraction, $V_f$, in the loading direction was equal to 0.70%.

### 2.1.4. Matrix

For the matrix, a commercial grout [29] was selected for its low viscosity before hardening to allow a manufacturing process by pouring. The mixture consisted of ordinary Portland cement, siliceous sand, and admixtures. Additionally, this grout was characterized by its compensated shrinkage and relatively small aggregate size, which allowed for optimal flow through the textiles. Table 2 summarizes the main properties of the matrix.

**Table 2.** Properties of the cementitious matrix, based on [30].

| Aggregate Size (mm) | Compressive Strength (28 d) (MPa) | Flexural Tensile Strength (28 d) (MPa) | Density after Mixing (kg/m³) | Young's Modulus (GPa) | Water/Mortar Ratio (-) |
|---|---|---|---|---|---|
| 0–1.6 | 70 | 12 | 2010 | 9 | 0.15 |

## 2.2. TRC Manufacturing Process

All TRC beams were manufactured in individual 1200 mm × 60 mm × 15 mm molds (Figure 5a). First, an oil film was applied at the bottom of the mold to facilitate the demolding process. The textile reinforcement was then placed in the mold and the matrix was poured. After filling the mold with mortar, the surface was covered with plastic film and the molds were sealed for 28 days to cure at room temperature. Six specimens were manufactured for each configuration. After demolding, the specimens' side surface was painted white and speckled to allow the DIC monitoring of displacements and cracks during the experimental campaign. Figure 5b shows an image of the prepared spacer configuration before casting, and Figure 5c shows the TRC beams after demolding.

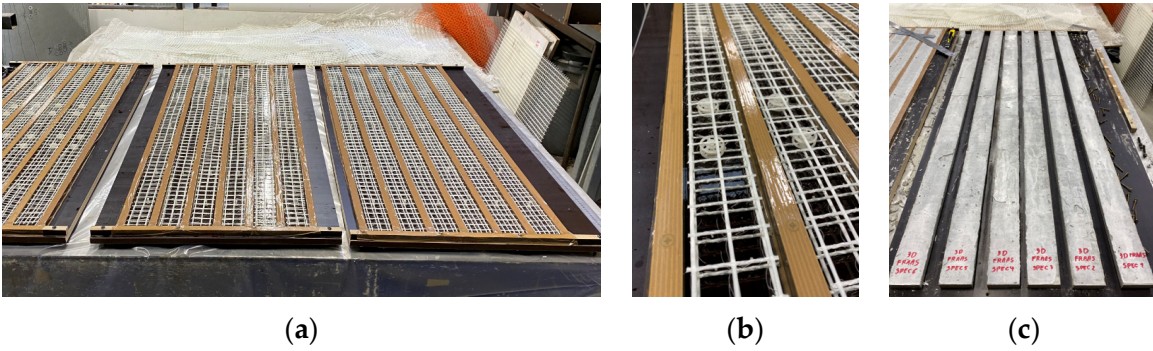

| (a) | (b) | (c) |

**Figure 5.** The specimen manufacturing process: (**a**) view of the molds with textiles; (**b**) zoom with visible connections; (**c**) specimens after demolding.

## 2.3. Experimental Test Setup

The four-point bending test setup is shown in Figure 6. The span was equal to 550 mm, and the distance between the loading pins was 100 mm. The specimens were loaded using an Instron 5885H universal electromechanical test bench with a crosshead rate of 2 mm/min. Two DIC cameras were mounted to monitor the side of the specimens and allow a full-field representation of the vertical displacement (Figure 6b). The pixel density of the cameras was 2448 pixels by 2048 pixels. To allow for the highest accuracy in the strain measurements, the highest pixel density was selected in the span direction (longitudinally to the sample). The subset size, step size, and strain window size were selected as 21, 7, and 9, respectively. The speckle size varied between 1.75 mm and 2 mm and the resolution was 0.23 mm/pixel.

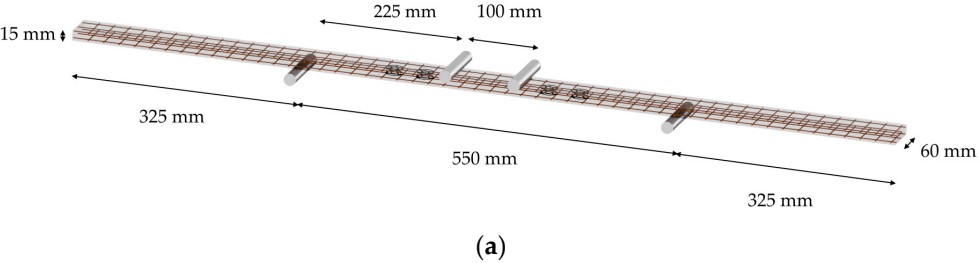

(**a**)

**Figure 6.** *Cont.*

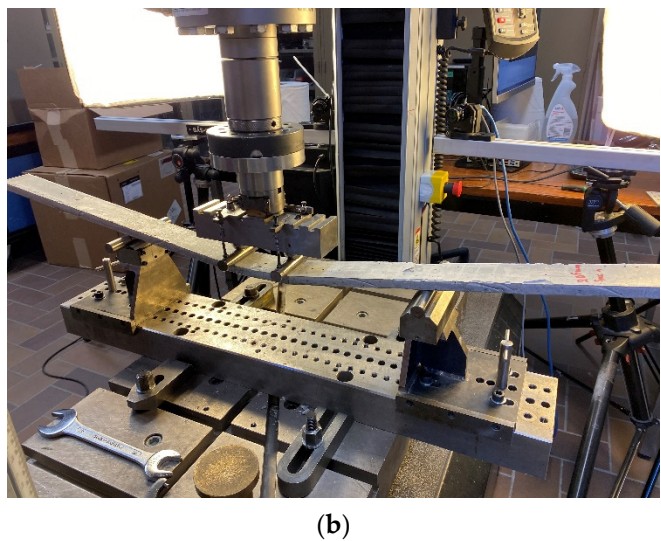

(**b**)

**Figure 6.** Four-point bending test setup: (**a**) schematic of the experimental setup; (**b**) picture of a loaded beam with DIC cameras.

## 3. Results and Discussion

The force-displacement curves of all six tested samples for each of the eight configurations are given in Figure 7a–h. Figure 8 presents a comparative overview of the representative samples for each configuration. An overview of the post-cracking bending stiffness ($K_3$, calculated between 20 mm and 35 mm vertical displacement), the modulus of rupture (MOR), and the cracking parameters are reported in Table 3. The crack analysis was performed at a fixed vertical displacement of 40 mm for each sample. All values in Table 3 are reported, with their respective standard deviations.

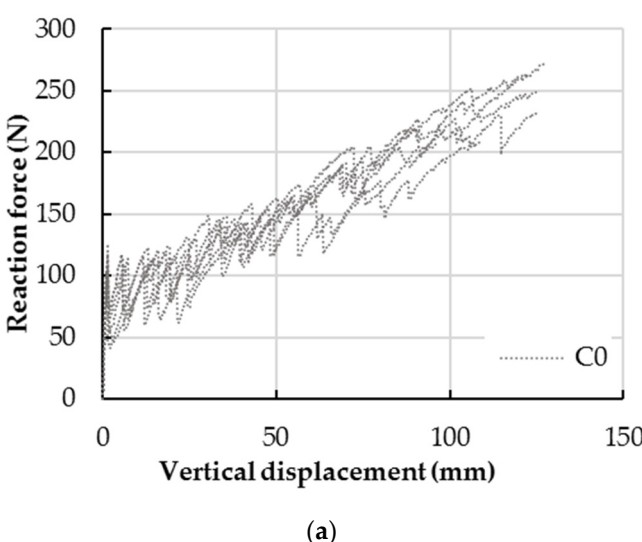

(**a**)

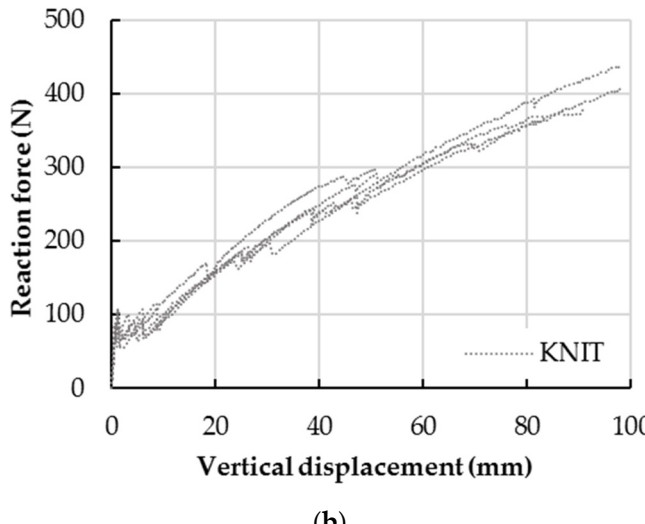

(**b**)

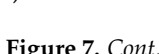
**Figure 7.** *Cont.*

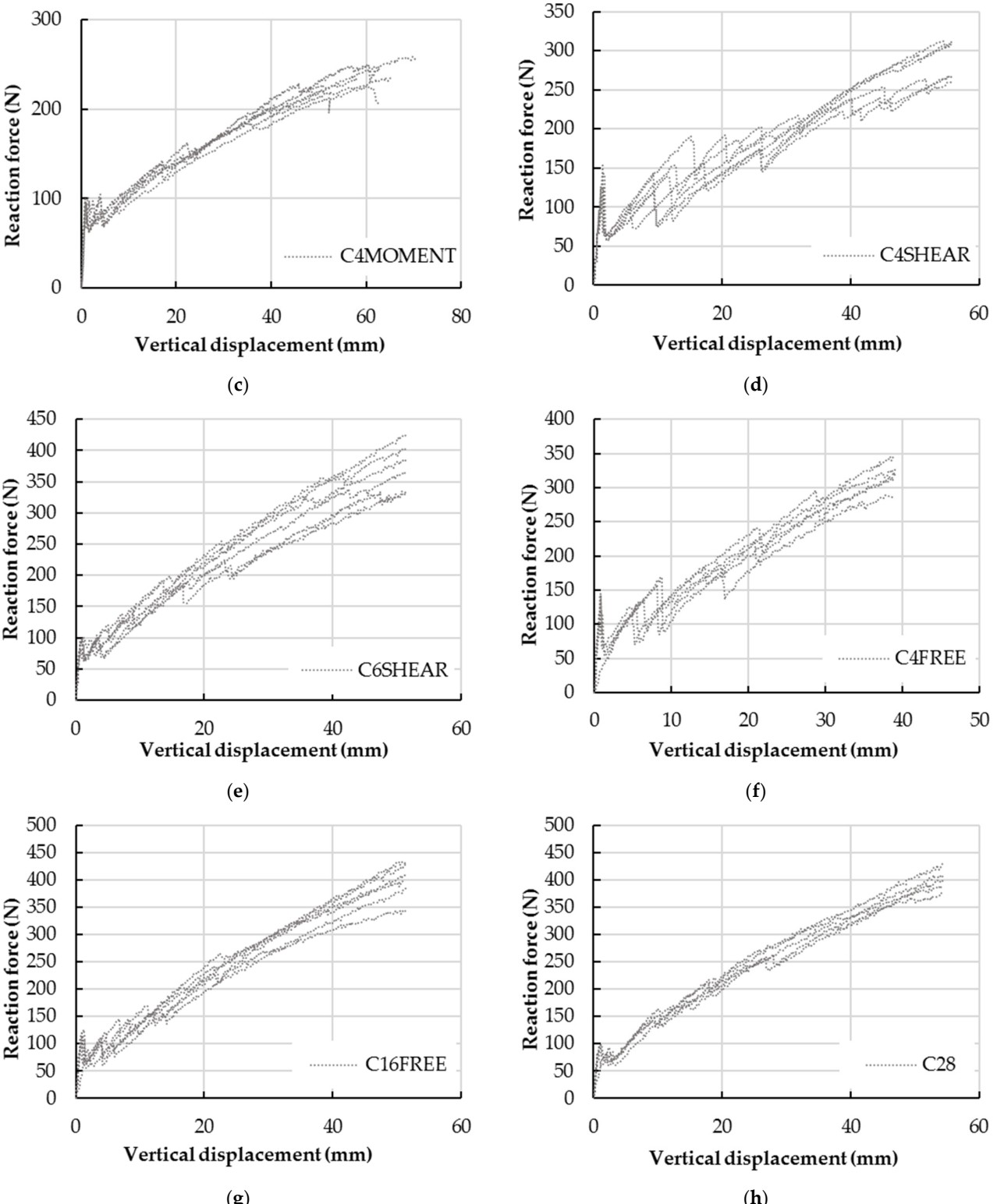

**Figure 7.** Reaction force (N)-displacement (mm) curves for all tested samples: (**a**) C0: samples without transversal connections; (**b**) KNIT: samples with knitted transversal connections; (**c**): C4MOMENT: samples with 4 transversal connections, placed in the moment zone; (**d**): C4SHEAR: samples with 4 transversal connections, placed in the shear zone; (**e**): C6SHEAR: samples with 6 transversal connections, placed in the shear zone; (**f**) C4FREE: samples with 4 transversal connections, placed in the free end zone; (**g**) C16FREE: samples with 16 transversal connections, placed in the free end zone; (**h**) C28: samples with 28 connections placed over the entire length.

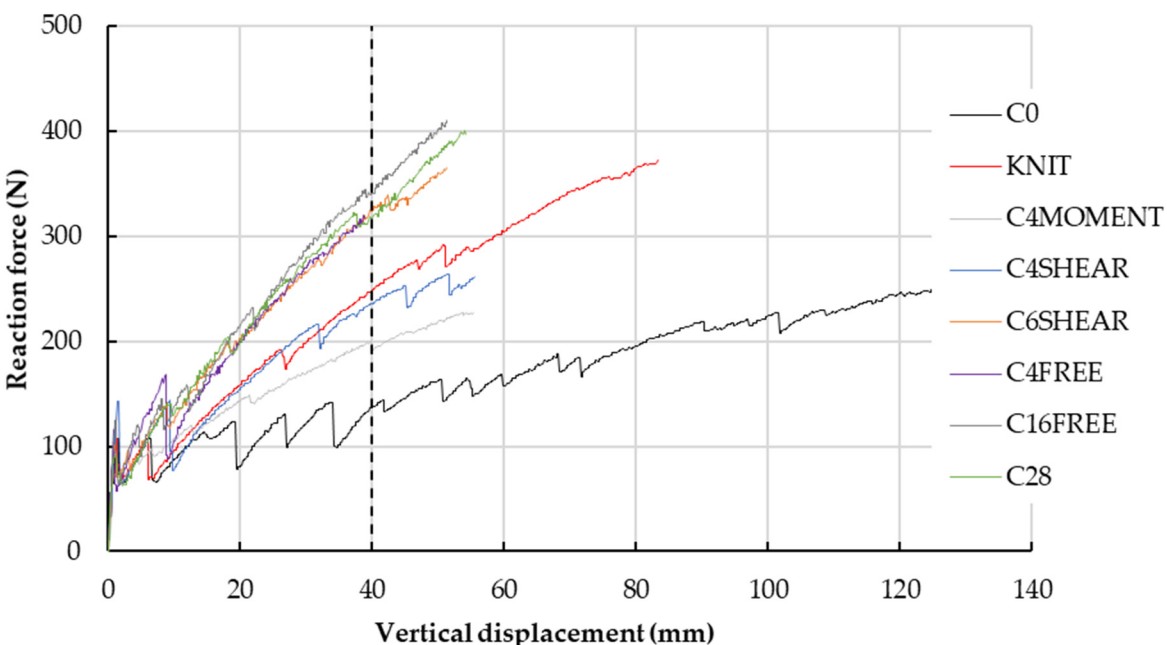

**Figure 8.** Comparison of the load-bearing behavior of the representative sample for each spacer configuration, demonstrating the positive influence of the spacer connections. The dashed line indicates the displacement level (40 mm) used for the DIC analysis.

**Table 3.** Summary of experimental results for the considered configurations.

| Conf. | $K_3$ (N/mm) | MOR (MPa) | Cracking Data at a Vertical Displacement of 40 mm | | | | | |
|---|---|---|---|---|---|---|---|---|
| | | | # Cracks Total | Crack Width Total (mm) | Crack Spacing Total (mm) | # Cracks Mid | Crack Width Mid (mm) | Crack Spacing Mid (mm) |
| C0 | $1.74 \pm 0.78$ | $11.9 \pm 1.2$ | 6 | $0.39 \pm 0.16$ | $46 \pm 13$ | 2 | $0.56 \pm 0.11$ | $62.0 \pm 0.0$ |
| KNIT | $4.40 \pm 0.72$ | $19.7 \pm 2.7$ | 7 | $0.38 \pm 0.12$ | $36 \pm 11$ | 2 | $0.35 \pm 0.10$ | $40.31 \pm 0.18$ |
| C4MOMENT | $2.82 \pm 0.52$ | $11.93 \pm 0.60$ | 8 | $0.52 \pm 0.29$ | $21.7 \pm 1.9$ | 6 | $0.47 \pm 0.18$ | $29.8 \pm 8.0$ |
| C4SHEAR | $3.98 \pm 0.78$ | $17.5 \pm 3.6$ | 5 | $0.41 \pm 0.23$ | $58 \pm 20$ | 3 | $0.64 \pm 0.15$ | $40.1 \pm 2.0$ |
| C6SHEAR | $6.5 \pm 1.9$ | $20.8 \pm 3.4$ | 4 | $0.68 \pm 0.28$ | $35 \pm 12$ | 4 | $0.68 \pm 0.28$ | $35 \pm 12$ |
| C4FREE | $7.0 \pm 2.3$ | $16.3 \pm 6.3$ | 2 | $1.608 \pm 0.025$ | $100 \pm 0.0$ | 2 | $1.608 \pm 0.025$ | $100 \pm 0.0$ |
| C16FREE | $7.1 \pm 1.5$ | $21.1 \pm 2.3$ | 3 | $0.92 \pm 0.35$ | $62.8 \pm 7.1$ | 3 | $0.92 \pm 0.35$ | $62.8 \pm 7.1$ |
| C28 | $6.7 \pm 2.1$ | $20.7 \pm 2.4$ | 6 | $0.41 \pm 0.17$ | $26 \pm 13$ | 4 | $0.30 \pm 0.11$ | $19.98 \pm 0.31$ |

$K_3$ was obtained as the slope of the load-displacement curve:

$$K_3 = \frac{F_{35} - F_{20}}{35 \text{ mm} - 20 \text{ mm}} \qquad (1)$$

where:

$F_{20/35}$: Reaction force (N) at 20 and 35 mm displacement, respectively.
The MOR was obtained using:

$$MOR = \frac{3 \, F \, a}{b \, h^2} \qquad (2)$$

where:

F: Reaction force (N);
a: Distance between supports and loading pins (225 mm);
b: Width of the sample (60 mm);

h: Height of the sample (15 mm).

In general, the specimens of each configuration yielded repeatable results. It should, however, be noted that, due to the nature of the tests (four-point bending), the crack parameters obtained over the total span of the beams were subjected to relatively high standard deviations (Table 3) since, by default, the cracks in the constant moment zone were larger than the cracks in the shear zone of the sample. Therefore, the cracking parameters and their standard deviations were reported in two ways: (i) data averaged over the entire TRC beam, and (ii) data averaged only over the constant moment zone, between the loading pins (Table 3).

The global macroscopic behavior of all the specimens followed the typical three-stage response of TRCs, including (i) the first pre-cracking stage, dominated by mortar properties, (ii) a multiple cracking stage, and (iii) a post-cracking stage, dominated by the fiber properties and textile anchorage [3]. At the level of the post-cracking bending stiffness, clear differences were observed between the different configurations, containing the same in-plane textiles but with spacers at different locations. Since the transversal connections did not affect the in-plane fiber volume fraction (unidirectional $V_f$ = 0.70% for all configurations), these differences could be attributed to changes in the anchorage of the in-plane textiles via the bridging provided by the connections, as will be discussed in more depth in the following sections. All the specimens failed due to fiber pull-out, however, depending on the presence of transversal connections, the pull-out happened at different load levels.

Figure 7a shows the experimental results of the six samples of the C0 configuration, without transversal connections (see Figure 4 for an overview of all the configurations). This configuration exhibited the lowest flexural post-cracking bending stiffness (1.74 ± 0.78 N/mm) and MOR (11.9 ± 1.2 MPa) out of all the configurations (see Figure 8 and Table 3 for comparison). These observations confirm the previous results published in the literature, namely, that 2D TRC composites exhibit lower post-cracking stiffness in flexural loading conditions, compared to 3D TRC composites [20–23]. The macroscopic mechanical behavior obtained from the C0 configuration will henceforth be considered representative of the in-plane reinforcement alone, without any transversal connections. In this reference configuration, the anchorage of the loaded textile yarns was achieved only through their bond with the cementitious matrix, as well as by the presence of the perpendicular yarns in the 2D planar textiles.

The four-point bending results of the six samples of the knitted 3D textile configuration (KNIT) are given in Figure 7b. Compared to the reference (C0), this configuration exhibited a considerable increase in post-cracking bending stiffness ($K_3$ = 4.40 ± 0.72 N/mm) and MOR (19.7 ± 2.7 MPa). A reduction in crack width was also observed, especially in the mid-section between the loading pins, at 0.35 ± 0.10 mm, compared to 0.56 ± 0.11 mm. These results confirmed the observations found in the literature regarding shorter beams, where 3D textiles with knitted connections exhibited a superior flexural response compared to their 2D-equivalent counterparts [22].

Figure 7c shows the experimental curves of the C4MOMENT configuration. The post-cracking bending stiffness (2.82 ± 0.52 N/mm) and MOR (11.93 ± 0.60 MPa) were both higher than for the reference C0 configuration but were lower than all other spacer TRC configurations (see Figure 8 and Table 3). At the level of crack formation, the C4MOMENT configuration resulted in multiple cracks, with relatively low crack widths. Compared to the other configurations with spacer connections, this alternative provided the second lowest crack width in the mid-span of the sample; only the C28 with 28 spacers distributed over the specimen length had lower crack widths. These results showed that when placed in the constant moment zone (between the loading pins), the spacer connections could improve the macroscopic flexural behavior of the TRC, but this was to a lower extent than in the KNIT configuration.

Figure 7d,e presents the experimental results of the six samples of the C4SHEAR and C6SHEAR configurations, respectively, where the spacer connections were concentrated

between the supports and loading pins. The C4SHEAR configuration exhibited a similar post-cracking bending stiffness ($K_3$ = 3.98 ± 0.78 N/mm) and MOR (17.5 ± 3.6 MPa) as the KNIT (see Table 3), while the C6SHEAR configuration exhibited an even higher post-cracking bending stiffness ($K_3$ = 6.5 ± 1.9 N/mm) and MOR (20.8 ± 3.4 MPa). These results showed that, when placed in the shear zone of the flexural setup, spacer connections were able to replicate (C4SHEAR) or even exceed (C6SHEAR) the anchorage provided by knitted transversal connections in TRCs (KNIT). Both the C4SHEAR and C6SHEAR configurations exhibited more and larger crack widths compared to the KNIT configuration; the crack width in the mid-section was almost doubled. On the other hand, the total number of cracks in these two configurations remained lower than in the KNIT configuration (see Table 3).

Figure 7f,g shows the experimental curves of the six samples of the C4FREE and C16FREE configurations. Both configurations exhibited higher post-cracking bending stiffness than the KNIT configuration, which was 7.0 ± 2.3 N/mm and 7.1 ± 1.5 N/mm, respectively. The C4FREE configuration, when compared to the KNIT configuration, had a relatively low MOR of 16.3 ± 6.3 MPa, as well as large crack widths of 1.608 ± 0.025 mm. The C16FREE configuration had a MOR of 21.1 ± 2.3 MPa and crack widths in the mid-span of 0.92 ± 0.35 mm. The combination of high post-cracking bending stiffness and large crack openings suggests that placing the spacer connections in the outer (free) zone of the samples improved the anchorage of the in-plane textiles the most notably, but, at the same time, could not support the crack formation in the central part of the beams due to the lack of connections there.

Figure 7h shows the experimental curves of the C28 configuration. This configuration was characterized by high values of post-cracking bending stiffness ($K_3$ = 6.7 ± 2.1 N/mm) and MOR (20.7 ± 2.4 MPa), as well as the lowest crack widths in the mid-section of the beams (0.30 ± 0.11 mm). At the level of post-cracking bending stiffness and MOR, the C6SHEAR, C16FREE, and C28 configurations exhibited comparable results. This suggests that the anchorage improvement provided by adding more spacer connections did not continuously increase, but instead reached a saturation value.

Summarizing the aforementioned trends, the experimental results suggest that spacer connections are capable of providing mechanical anchorage to the in-plane textiles and that the degree of anchorage is dependent on their positioning. The closer the spacer connections are placed toward the outer ends of the beam, the higher the increase in post-cracking bending stiffness. At the same time, the closer the spacer connections are placed to the center of the beam, the more controlled the crack formation will be. While these spacer connections were originally developed solely to facilitate manufacturing, these research results thus highlight a new and original insight, namely, that they also provide significant added mechanical value. Their impact is shown, unexpectedly, to be highly significant. The increase in post-cracking bending stiffness obtained from the spacer connections in this study, in some cases even surpassed the values found for knitted connections; the $K_3$ of C6SHEAR, C4FREE, C16FREE, and C28 surpassed the value of 4.40 N/mm observed for the KNIT configuration (see Table 3).

It is believed by the authors that the spacer connections anchor the in-plane textiles by providing a local bridge between the top and bottom textile layers. As a result, in loading conditions where the relative displacement of the two connected textiles is opposite to each other (such as in bending, see Figure 9), the longitudinal elongation and shortening of the bottom and top reinforcement layers, respectively, are hindered by the connections' embedment within the matrix. This hypothesis endorses the experimental findings since the respective textile layers are subjected to higher relative displacement the further away from the center of the beam they are located. The authors believe that the bridge-anchorage effect provided by spacer connections is distinct from the traditionally recognized anchorage mechanisms of textiles in mortar, such as (i) the mechanical anchorage of the textile reinforcement grid within the mortar, (ii) the bond at the interface between the outer circumference of the longitudinal rovings and the mortar, and (iii) the inter-fiber friction

between the filaments within the rovings [1,10,11,31]. In this case, the bridging effect and connections take place between two separate textile reinforcement layers that are subjected to different loading conditions.

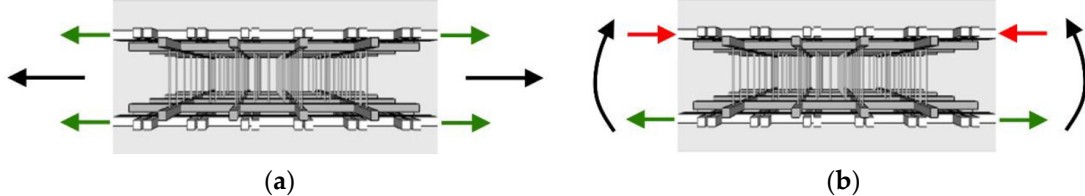

**Figure 9.** Evaluation of textile displacement under (**a**) tensile (green tensile stresses in top and bottom region) and (**b**) flexural loading conditions (green tensile stresses in bottom region and red compressive stresses in top region).

## 4. Conclusions

This research evaluated the potential influence of the number and position of spacer connections on the flexural behavior of TRC composites. This research is unprecedented as these spacer connections were originally destined to ease the manufacturing process of TRC composites and to tightly control the location of the textiles within the concrete cross-section. A flexural experimental campaign was conducted on eight different configurations (six samples per configuration), with one 2D equivalent configuration, one 3D knitted configuration, and six configurations with spacer connections.

The experimental results showed that both the position and the number of spacer connections within the TRC beam's length influenced the amount of anchorage that they provided. The influences were quantified at the level of the post-cracking bending stiffness $K_3$, MOR, and crack formation. The general trend was that a higher amount, as well as a position closer to the outer, free zone of the sample, resulted in increased values of $K_3$ and MOR. At the same time, placing the spacer connections closer to the specimen's center improved the crack formation considerably. With the correct placement of the spacer connections, an anchorage that was similar to or even better than that achieved with integrated, knitted connections was obtained. At a certain anchorage level, increasing the number of connections no longer resulted in improved mechanical properties, hinting toward its reaching the saturation level of the spacer anchorage mechanism.

A hypothesis was provided for the local anchorage mechanism provided by the spacers: these connections form local bridges between the top and bottom textile layers. Within the matrix, this bridge is cemented in place, which hinders the (longitudinal) relative displacement of these in-plane textiles. Hence, this anchorage effect is believed to be more effective toward the supports and the edges of the samples, where the relative displacement between the top and bottom textile layers is larger, as was observed experimentally.

These experimental results offer supporting evidence that spacer connections can have a significant mechanical benefit that is currently not considered in TRC design, even though their (original) purpose is to enable textile spacing during manufacturing. By correctly tuning the amount and position of the transversal connections in TRC applications, these mechanical properties can be significantly improved. Future research should investigate the anchorage mechanism of these connections in more complex loading conditions, such as with (bi-directional) plates and curved TRC elements.

**Author Contributions:** Conceptualization, M.E.K., D.V.H. and T.T.; methodology, M.E.K. and T.T.; validation, M.E.K. and T.T.; formal analysis, M.E.K.; investigation, M.E.K.; resources, D.V.H. and T.T.; data curation, M.E.K.; writing—original draft preparation, M.E.K.; writing—review and editing, D.V.H. and T.T.; supervision, T.T.; project administration, D.V.H. and T.T.; funding acquisition, D.V.H. and T.T. All authors have read and agreed to the published version of the manuscript.

**Funding:** Financial support given by the Research Foundation–Flanders (FWO-Vlaanderen) via project number G030721N is gratefully acknowledged.

**Institutional Review Board Statement:** Not applicable.

**Informed Consent Statement:** Not applicable.

**Data Availability Statement:** The data can be requested from the authors.

**Acknowledgments:** The authors deeply acknowledge the financial support from FWO, as reported above as well as the input from all the material providers during this study.

**Conflicts of Interest:** The authors declare no conflict of interest.

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
