# Peer review of "Improving the Anchorage in Textile Reinforced Cement Composites by 3D Spacer Connections: Experimental Study of Flexural and Cracking Behaviors"

_jcs, doi:10.3390/jcs6120357_

Round 1
Reviewer 1 Report
This is very well written manuscript. Background/introduction is sufficient enough. Design of experiment is logical and clear. Conclusions are reasonable based on the presented results. I do not have any questions on this manuscript. I think it is suitable for J. Compos. Sci. scope and meet journal's standard.
This manuscript describes a new way to improving the anchorage in TRC composite, using 3D spacer. In the introduction part, authors well explained that textile reinforce cement is usually reinforced by planar textile (2D structure). Transversal connection in 3D TRC has been reported in the literature, but It is currently clear if or not these individual spacer connections can achieve any level of anchorage improvement.
In the result and discussion section, author clearly evaluated the influence of spacer and position of spacer. For example, in Figure 7 and 8, it shows that 3D TRC exhibits higher post cracking stiffness and the degree of anchorage is dependent on their positioning. This research provide insight into how to improve mechanical performance while keeping the manufacturing process of TRC composites easy.
Lastly, authors proposed hypothesis for this improvement. These 3D connectors generate local bridges between top and bottom, which hinders the longitudinal displacement.
Overall, this manuscript is well written and logical. Results support their hypothesis and conclusions. Thus, I propose to accept this manuscript in present form.
Author Response
Dear Reviewer,
In name of all the authors, I would like to thank you for taking the time to review our manuscript as well as the recommendation for publication.
We will perform the minor adaptations proposed by Reviewer 3 and resubmit the manuscript.
Best regards,
Michael El Kadi
Reviewer 2 Report
I recommend publishing
Author Response

(The authors gave the same response as above.)

Reviewer 3 Report
* This paper presents interesting results focused of rupture properties throughout flexural tests.
* Page 5, line 127: May be the authors should revise the pixel density, since the first figure may be erroneous: "The pixel density of the cameras was 2448 pixels by 2048 pixels.". Wouldn't it be "2048" instead of "2448" ?
* Page 5, line 135: Please define "post-cracking bending stiffness K3" at its first occurence. Please provide some details on the way it is calculated, with a formula.
* It would also have been very interesting to provide also elastic properties, i.e. Young modulus and Poisson coefficient.
* Please define the acronym MOR as "Modulus of Rupture (MOR)" at the begining of the paper, i.e. in the abstract.
Author Response
Dear Reviewer,
In name of all the authors, I would like to thank you for taking the time to review our manuscript and provide your valuable feedback, please find attached a file addressing all your comments.
Best regards,
Michael El Kadi

Round 2
Reviewer 3 Report
All the remarks and comments have been taken into account. When necessary appropriate answers have been provided by the authors.